# *Clostridioides difficile* in South American Camelids in Germany: First Insights into Molecular and Genetic Characteristics and Antimicrobial Resistance

**DOI:** 10.3390/antibiotics12010086

**Published:** 2023-01-04

**Authors:** Ines Dost, Mostafa Abdel-Glil, Gernot Schmoock, Christian Menge, Christian Berens, Belén González-Santamarina, Elisabeth Wiegand, Heinrich Neubauer, Stefan Schwarz, Christian Seyboldt

**Affiliations:** 1Institute of Bacterial Infections and Zoonoses, Friedrich-Loeffler-Institut, Federal Research Institute for Animal Health, Naumburger Straße 96a, 07743 Jena, Germany; 2Institute of Molecular Pathogenesis, Friedrich-Loeffler-Institut, Federal Research Institute for Animal Health, Naumburger Straße 96a, 07743 Jena, Germany; 3Institute of Microbiology and Epizootics, Centre for Infection Medicine, Department of Veterinary Medicine, Freie Universität Berlin, 14163 Berlin, Germany; 4Veterinary Centre for Resistance Research (TZR), Freie Universität Berlin, 14163 Berlin, Germany

**Keywords:** *Clostridioides difficile*, South American camelids, antimicrobial resistance, whole genome sequencing, RT 002/2, RT 015, RT 029, RT 078, RT AI-75

## Abstract

Little is known about zoonotic pathogens and their antimicrobial resistance in South American camelids (SAC) in Germany including *Clostridioides* (*C*.) *difficile*. The aim of this study was to investigate prevalence, molecular characteristics and antimicrobial resistance of *C. difficile* in SAC. Composite SAC faecal samples were collected in 43 husbandries in Central Germany and cultured for *C. difficile*. Toxinotyping and ribotyping was done by PCR. Whole genome sequencing was performed with Illumina^®^ Miseq™. The genomes were screened for antimicrobial resistance determinants. Genetic relatedness of the isolates was investigated using core genome multi locus sequence typing (cgMLST) and single nucleotide polymorphism analysis. Antimicrobial susceptibility testing was done using the Etest^®^ method. Eight *C. difficile* isolates were recovered from seven farms. The isolates belonged to different PCR ribotypes. All isolates were toxinogenic. cgMLST revealed a cluster containing isolates recovered from different farms. Seven isolates showed similar resistance gene patterns. Different phenotypic resistance patterns were found. Agreement between phenotypic and genotypic resistance was identified only in some cases. Consequently, SAC may act as a reservoir for *C. difficile*. Thus, SAC may pose a risk regarding zoonotic transmission of toxinogenic, potentially human-pathogenic and resistant *C. difficile* isolates.

## 1. Introduction

The Gram-positive, anaerobic, spore-forming bacterium *Clostridioides* (*C.*) *difficile* is a well-known enteric pathogen. It can cause diseases in humans and animals ranging from mild diarrhoea to sudden death, but asymptomatic carriage is also possible [1,2].

*C. difficile* infection (CDI) was initially recognized as a mainly nosocomial disease in humans, but the number of community-associated *C. difficile* infections (CA-CDI) has risen in the last years representing approximately 33 to 41% of the CDI cases in the USA and possibly up to 25% in Europe [3,4,5]. In many CA-CDI cases, well-known traditional risk factors, such as antimicrobial treatment or contact with health-care settings, are absent [5].

*C. difficile* has been isolated from many animal species and environmental niches [1,6]. Although a zoonotic transmission has not been experimentally proven so far, genetically indistinguishable *C. difficile* isolates found in pigs and humans strongly suggest zoonotic transmission [7,8].

A risk for successful CDI treatment is the emergence of antimicrobial resistance. *C. difficile* isolates show resistance to various antimicrobial agents and can serve as antimicrobial resistance reservoir for other Gram-positive bacteria due to many mobile genetic elements in their genome [9,10].

During the last years, the number of South American camelid (SAC) husbandries in Germany has steadily increased. According to a survey conducted among German SAC owners (255 participants), more than every second owner (55.1%) started keeping these animals from 2014 to 2019 [11]. In August 2020, 12,458 SAC were registered on the website of the “Alpaka Zucht Verband Deutschland e.V.” (Alpaca Breeding Association Germany) [11], this number doubled to approximately 25,000 animals in October 2022 (https://www.azvd.de/; accessed on 11 October 2022). Llamas and alpacas, mostly kept for hobby purposes, are used for wool production, breeding, trekking tours, animal assisted therapy or landscape conservation [11,12]. This increasing popularity and their frequent close contact with humans urge a risk analysis on potential zoonotic pathogens. 

The aim of this study was to investigate the prevalence, molecular characteristics and antimicrobial resistances of *C. difficile* recovered from SAC.

## 2. Results

### 2.1. Prevalence and Molecular Characteristics

Eight samples from seven husbandries were positive for *C. difficile* (typical phenotype and confirmation by PCR of *cdd3*) resulting in an overall prevalence of 16% (7/43) at the husbandry-level. All isolates were obtained only from enrichment cultures and belonged to five different PCR ribotypes (RT): RT 002/2 (3/8), RT 015 (2/8), RT 029 (1/8), RT 078 (1/8) and RT AI-75 (1/8). All eight isolates were positive for the toxin genes *tcdA* and *tcdB*. Only the isolate 19S0136 (RT 078) was positive for the binary toxin genes *cdtA* and *cdtB* (Table 1).

### 2.2. Whole Genome Sequencing (WGS)

Quality control values collected during WGS data analysis by WGSBAC revealed that all isolates belonged to *C. difficile* [13]. The genome assembly sizes ranged from 4 to 4.5 Mbp and theoretical coverages from 58- to 98-fold. Values for the average nucleotide identity (ANI) between all genomes were above 96%.

The analysis of single nucleotide polymorphisms (SNPs) is shown in Table 2. The isolates 19S0260 and 19S0262 were identical (no SNP). They possessed five SNPs difference to 19S0264, and 19S0105, another sequence type (ST) 8 isolate, showed 116—121 SNPs to the RT 002/2 isolates (19S0260, 19S0262, 19S0264). The two RT 015 (ST 10) isolates 19S0160 and 19S0161 varied by six SNPs. 19S0136 (RT 078) was the most distant isolate with more than 91,000 SNPs difference to every other isolate.

A total of 2147 core genes were compared by core genome multi locus sequence typing (cgMLST) [14]. The cgMLST revealed four different clusters in accordance with the ST with a cluster type threshold of three differing alleles [15]. According to cgMLST, the RT 002/2 isolates 19S0260 and 19S0264 were identical and revealed one allele difference to 19S0262. 19S0160 and 19S0161 (both RT 015) also differed in one allele (Figure 1).

Seven isolates showed a similar content of resistance genes, including *bla*_CDD_-1, *vanZ1* and the *vanG* gene cluster, consisting of the regulatory genes *vanR* and *vanS*, as well as the effector genes *vanG* and *vanT.* The isolate 19S0136 (RT 078) possessed *aadE*, *bla*_CDD_-1, *tet*(40), *tet*(M) and *vanZ1* (Table 3). In addition, this isolate showed nucleotide changes in the genes *gyrA* and *gyrB*, which resulted in several amino acid substitutions in the corresponding proteins GyrA and GyrB, some of them previously reported to confer fluoroquinolone resistance [10]. In GyrA, two amino acid substitutions were found (Lys-413-Asn and Thr-82-Val), whereas three substitutions were present in GyrB (Gln-160-His, Ser-416-Ala and Ser-366-Val). The substitution Val-130-Ile in GyrB was found in both RT 015-isolates, 19S0160 and 19S0161 (Table 4).

### 2.3. Antimicrobial Susceptibility Testing (AST)

The AST results are summarised in Table 5. All isolates were susceptible to metronidazole (MEZ), vancomycin (VAN), meropenem (MEP) and amoxicillin-clavulanate (AMC). Seven isolates were susceptible to moxifloxacin (MOX), tetracycline (TET) and classified as wild type (WT) with respect to erythromycin (ERY). In contrast, isolate 19S0136 was resistant to MOX and TET and classified as ERY-non wild type (NWT). Four isolates were classified as resistant to penicillin (PEN) and ampicillin (AMP), two isolates were classified as intermediate for both antimicrobial agents (19S0160 and 19S0264). Isolate 19S0136 was resistant to PEN and intermediate for AMP, while isolate 19S0161 was resistant to AMP and intermediate for PEN. For chloramphenicol (CHL), seven isolates were classified as susceptible, while isolate 19S0105 was classified as CHL-intermediate. Seven isolates were resistant to clindamycin (CLI), 19S0136 was classified as CLI-susceptible. All isolates were classified as NWT for ciprofloxacin (CIP) with minimum inhibitory concentration (MIC) values of >32 mg/L. For linezolid (LZD), five isolates had MIC values of 3 mg/L, whereas isolates 19S0161, 19S0160 and 19S0266 showed MIC values of 4 mg/L, 6 mg/L and 8 mg/L, respectively.

## 3. Discussion

### 3.1. Clostridioides difficile and (South American) Camelids in Literature—A Short Review

As only a few samples were investigated in this study, a literature search was performed to gain a broader overview of *C. difficile* in SAC.

For this, the literature database PubMed was searched with the term *((Clostridium difficile) OR (Clostridioides difficile)) AND ((camel*) OR (llama) OR (alpaca))*. A total of 24 results were obtained (22 December 2022). Of these 24 results, 21 articles had to be excluded as they investigated another topic. Mostly, they were related to single-domain camelid antibodies as a therapeutic option for CDI (13 articles). Three articles investigated *C. difficile* and its prevalence in camels [21,22,23]. *C. difficile* was found in two of these studies, none of these research articles described *C. difficile* in SAC. 

In one study from Saudi Arabia, camel minced meat was investigated. *C. difficile* was found in 4% (4/100) of the samples. All isolates were positive for toxin B, three isolates were positive for toxin A. AST was performed by broth microdilution for seven different antimicrobial agents (ceftriaxone, TET, CLI, MEZ, PEN, MOX and AMC). The isolates showed different antimicrobial resistance patterns (TET, CLI + TET, ceftriaxone + TET, PEN + ceftriaxone + MOX) [21]. 

In another study, *C. difficile* isolates were found in camel faeces (in 1 of 25 samples; the isolate belonged to RT IR43) and on a postevisceration camel carcass (25 carcasses were tested; the isolate belonged to RT IR48) in Iran. No further characterisation of these isolates was performed [22].

The third study investigated 124 camel meat samples, all of them were negative for *C. difficile* [23].

Next to these articles, a fourth article was found, which investigated different samples for *C. difficile* including one llama faecal sample. This sample was negative for *C. difficile* [24]. 

The lack of investigations of *C. difficile* in camelids shows the importance of this study and that more research is needed regarding *C. difficile* in SAC. This is the first study to report the presence of *C. difficile* in SAC.

### 3.2. Prevalence of Clostridioides difficile in German South American Camelids

Comparing the prevalence of 16% at the husbandry-level to the prevalence of *C. difficile* of other herbivore livestock-farms, this prevalence is similar to that for calves (19.8% of 101 Italian farms) but lower than that of Slovenian dairy-farms with an average of 39.8% [25,26]. Taking these results into consideration, *C. difficile* seems to be as present in SAC-husbandries as in other herbivore livestock-husbandries, but with an apparently lower prevalence.

All eight isolates were cultivated from enrichment cultures, the bacterial or spore load of the samples was presumably not sufficiently high for detection in direct cultures. This is not surprising considering that pooled faecal samples of animals of presumably different age classes were investigated. The prevalence of *C. difficile* is known to be age-related and is higher in younger individuals in many animal species and in humans [1,5,27]. The calculation of a prevalence at the animal-level for these samples is not possible, because composite faecal samples were used which had been collected in a previous study [12]. However, considering that 9% (8/94) of the samples were tested positive with a low bacterial (or spore) load, the obtained results are comparable to the prevalence in small ruminants (2–9.5%) and adult horses (0–8.4%), which are probably the most comparable animal species regarding husbandry conditions and human—animal interactions [1]. 

### 3.3. Isolates of Different Toxinogenic Ribotypes Resistant to Antimicrobial Agents Were Detected

The five RTs found in this study have been detected in humans before [28,29], and as the corresponding isolates are positive for the toxin genes *tcdA* and *tcdB*, they may be capable of causing disease [30].

Overall, results obtained by cgMLST (Figure 1) correspond with results obtained by SNPs analysis (Table 2). The number of SNPs reflect clusters and differing alleles revealed by cgMLST with a close relatedness of isolates belonging to the same ST (0 to 121 SNPs). According to both methods, the most distant isolate was 19S0136 (RT 078).

With 19S0136 belonging to RT 078, a hypervirulent and multidrug-resistant isolate was found in our study [2,31]. RT 078 is livestock-associated and potentially zoonotic [1,7]. Our results from the AST (Table 5) and resistome data (Table 3 and Table 4) are in line with those of other studies, since resistance to TET, MOX and ERY is common in ST 11 and RT 078 isolates, respectively, as well as the presence of antimicrobial resistance genes for aminoglycosides and TET [31]. We could not identify the genetic basis for the high ERY MIC value as *erm* genes or other known macrolide resistance genes were not detected in the isolates studied. However, the absence of *erm* genes in ERY-resistant isolates is a common finding in *C. difficile* [10].

Three RT 002/2 isolates (19S0260, 19S0262, 19S0264) of ST 8 were found in different husbandries from Saxony-Anhalt. RT 002/2 is a sublineage of the emerging RT 002, which has been found in humans, animals and the environment [2]. Isolates of this RT can cause severe CDI and are associated with various antimicrobial resistances, e.g., resistances to fluoroquinolones and CLI [2]. For CIP and CLI, the three RT 002/2 isolates showed resistance; for MOX, the isolates were classified as susceptible. The RT 002/2 isolates found in this study seem to be closely related as revealed by cgMSLT as well as by SNP analysis. Interestingly, the classification as identical isolates differed for the two methods of analysis as depicted by the results for 19S0260, 19S0262 and 19S0264 in Figure 1 and Table 2. This can be explained by different approaches of the methods: with the SNP analysis, we looked for nucleotide substitutions, while insertions and deletions were ignored. SNPs analysis strongly depends on the reference sequence used as only similar regions are mapped and divergent parts of the genome are excluded from the analysis. It is possible that, using another reference strain, other regions would be in- or excluded from the analysis and slightly different results would be obtained as a consequence. In general, the closer related the reference strain is to the tested isolates, the less sequence elements are excluded from the analysis [32]. In contrast, cgMLST is a gene-by-gene approach. Predefined genes are selected and compared. The advantage of this method is the standardisation of nomenclature, thereby rendering a comparison of isolate data between laboratories much easier. In contrast to SNPs analysis, cgMLST ignores variations in the intergenic regions [33], and collapses genetic variations of genes into allelic numbers. Thus, cgMLST is considered less discriminatory in comparison to the core genome SNP (cgSNP) analysis methods. In our study, the well-characterized *C. difficile* strain 630 (GenBank accession numbers NC_009089 and NC_008226) served as reference genome for SNP calling. This genome was the first closed genome of the species and has been used as a seed genome for the development of the *C. difficile* cgMLST scheme [9,34]. In summary, the cgMLST based analysis supplemented with cgSNP calling and phylogenetic analysis are considered complementary methods for better delineation of the genomic relatedness of the isolates. Both methods resulted in similar genetic relatedness for the isolates investigated in this study. Comparing the results obtained by cgMLST with the results obtained by SNPs analysis, both methods are on a par and give comparable results.

Taking into consideration that the RT 002/2 isolates are closely related but originated from different husbandries, a common source and epidemiological connection seems possible. Perhaps the corresponding animals had direct contact, originated from the same breeding farm, received feed from the same source or had contact with the same persons. Unfortunately, this background information was not available due to the fact that the samples were collected in a previous study [12].

A fourth isolate belonging to ST 8 and RT AI-75 is 19S0105. This isolate showed similar results in AST compared to the RT 002/2 isolates, except for CHL. For CHL, 19S0105 was the only isolate classified as intermediate. Recently, two human RT AI-75 isolates found in Italy were classified as multidrug-resistant [28]. RT AI-75 has rarely been detected. For a better risk assessment of this RT, more data is needed.

A prevalent RT in Europe is RT 015 which is linked to CA-CDI and to animals [2]. The two isolates 19S0160 and 19S0161 found in this study belong to RT 015. They differed in one allele according to cgMLST, and six polymorphisms were detected in the SNPs analysis. This suggested a very close or clonal relatedness, given that allelic differences up to three in cgMLST can be regarded as clonal [15]. Interestingly, these isolates exhibited the same amino acid substitution Val-130-Ile in GyrB, which was not observed in any of the other isolates of this study. However, this substitution had also been identified in another study: a RT 015 isolate with an additional substitution in GyrA and two isolates (RT 020 and RT 011) harboured the substitution Val-130-Ile [35]. Like these RT 015 isolates, the isolates in the study of Mac Aogáin et al. showed MIC values > 32 mg/L for CIP and low MIC values for MOX [35]. However, it is uncertain whether the substitution Val-130-Ile confers a CIP-resistance phenotype, as all of our isolates showed MIC values > 32 mg/L for CIP regardless of their respective amino acid substitutions in Gyr proteins. The high MIC values for CIP of our isolates are in line with previous results, as CIP is one of the antimicrobial agents with the highest resistance rates measured for *C. difficile* [10]. In this regard, antimicrobial characteristics of *C. difficile* isolates from SAC correspond to *C. difficile* isolates from other sources. 

19S0266 belonged to RT 029. It showed slightly higher MIC values for penicillins than the other isolates. RT 029 is not as prevalent as RT 078 or RT 015 and has been detected in rodents from Dutch farms [36,37,38]. It is possible that this RT is present in farm- or stable-surroundings and that it has been transmitted either from the environment with vermin as potential vector or from humans to SAC.

In all isolates, the chromosomal class D β-lactamase gene *bla*_CDD_-1 was found which is common for *C. difficile* [10,39]. As in this study, reduced susceptibility for penicillins was observed in other studies (MIC ranges for AMP ≤ 0.5–4 mg/L and for PEN 0.5–2 mg/L) [25,40,41,42]. For AMC, an aminopenicillin combined with a β-lactamase inhibitor, and the carbapenem MEP, all isolates were classified as susceptible. In other studies, similar results were obtained with only a few isolates reported to be resistant [25,41,42,43]. 

In seven of the eight isolates, a *vanG* gene cluster (*vanG*, *vanR*, *vanS* and *vanT*) was found. All SAC isolates were susceptible to VAN (MIC range 0.75–1 mg/L). In previous studies, this cluster was detected in 85% (35/41) of clinical human isolates belonging to different RTs and in 69% (57/83) of isolates representing the main linages of *C. difficile*, respectively [44,45], but it did not confer resistance to VAN [45,46]. VAN MIC values of these SAC isolates underscore those findings showing that this cluster could be widely distributed in animal and human *C. difficile* isolates.

In summary, these SAC *C. difficile* isolates showed comparable antimicrobial resistance profiles as did isolates from other sources. Therefore, they may pose a similar risk to other putative *C. difficile* infection sources regarding the transmission of antimicrobial resistant isolates belonging to toxinogenic RTs.

### 3.4. South American Camelids and Antimicrobial Agents

The use of antimicrobial agents in SAC in Germany is restricted because the German animal health law lists camelids as livestock [47]. Therefore, only substances listed in the European Commission Regulation (EU) No 37/2010 for foodstuffs of animal origin can be used—including the following (antimicrobial) agents tested: AMP, PEN, AMC (clavulanic acid only bovine and porcine species), ERY and TET [48]. Exemptions can be made for “hobby-animals” which are not classified as food-producing animals. For these single cases, other substances can be used—but it is unknown how often this option is used for SAC. Next to legal limitations of antimicrobial use in SAC, there are no authorised antimicrobial drugs for these species in Germany. As a consequence, drugs with authorisation for small ruminants are reallocated and used [49].

It seems unlikely that the SAC *C. difficile* isolates developed resistance in their SAC hosts as the detected resistance patterns are in agreement with antimicrobial resistance characteristics of *C. difficile* in general [10], and particularly for the multidrug resistant 19S0136 isolate belonging to livestock-associated RT 078 [1,31]. Therefore, a transmission of already antimicrobial resistant *C. difficile* isolates to SAC is assumed.

### 3.5. Further Investigation Is Needed

With this investigation, first insights have been gained regarding *C. difficile* in SAC. However, there are still many open questions.

First, the question of how these SAC became infected or why these samples carried *C. difficile* isolates arises. One possibility could be a zoonotic transfer from humans to the SAC during direct contact while handling the animals. As mentioned above, all RTs detected in this study have been found in humans before [28,29], and zoonotic transmission has already been discussed in other studies [7,8]. Next to this transmission route, an infection or transmission via other livestock could have taken place. In total, 53.3% (136/255) of SAC owners also keep other animal species and 6.7% (17/255) even in direct contact with SAC [11]. Mostly, poultry is kept together with SAC [11]. *C. difficile* prevalences in healthy chickens range from 0–62% with a high isolate diversity [1]. Another possible infection source could have been the environment, the feed or vermin as *C. difficile* has been detected in water samples, compost, soil, sediment, vegetables or rodents and insectivores [5,6,38,41]. As composite faecal samples were collected from the pen floor or the meadow ground, a contamination of the samples, e.g., via airborne dispersal or vectors could have taken place [31]. However, this seems unlikely as highly similar RT 002/2 isolates (19S0260, 19S0262, 19S0264) were detected in samples obtained from different husbandries.

Another open question is whether *C. difficile* can cause disease in SAC. Due to pooled samples and the fact that no data concerning health or diarrhoeal status of the animals were available, it is not possible to give a statement about occurrence of clinical disease and economic relevance of *C. difficile* in SAC. Diarrhoea and colics can be induced by *C. difficile* [1]. These symptoms were reported by Neubert et al. to have been observed by 36.9% (94/255) and 15.3% (39/255) of SAC owners, respectively [11].

As only farms in Central Germany were sampled, this study can just give first insights in the prevalence and characteristics of *C. difficile* in SAC. Reported prevalence of *C. difficile* as well as the RT distribution in different geographical regions can be highly variable [1,2,5,31]. Therefore, further research is important—in Germany, but also in other regions.

Taking all findings together, toxinogenic, potentially human-pathogenic and antimicrobial resistant *C. difficile* isolates have been detected in SAC. Thus, it should be considered to implement a monitoring system for SAC—not only for *C. difficile* but also for other pathogenic and zoonotic bacteria and for antimicrobial resistance. This has been already recommended in a previous investigation of antimicrobial resistant *Escherichia coli* from SAC [50].

## 4. Materials and Methods

### 4.1. Sampling

In a pilot study on the prevalence of zoonotic and epizootic bacteria and antimicrobial-resistant *Escherichia coli* in SAC, 94 pooled faecal SAC samples were collected from 43 alpaca-/llama-husbandries in the German federal states of Saxony, Saxony-Anhalt and Thuringia between May and September 2019 [12]. The faecal samples were collected from the pen floor or from the meadow ground and were not allocated to individual animals. The number of samples per farm ranged from one to seven. In 41 (out of the 43) sampled farms, the number of animals (llamas and/or alpacas) ranged from 2 to 150, while data were not available for two farms.

### 4.2. Isolation of C. difficile

Two methods of isolating *C. difficile* were performed involving direct plating and pre-enrichment. For the enrichment, approximately 0.5 g of each faecal sample was inoculated in 10 mL *C. difficile* broth supplemented with CDMN selective supplement (Oxoid, Wesel, Germany) and 0.1% sodium-taurocholate. Aliquots of 100 µL of this mixture were plated onto CDMN agar (*Clostridium difficile* Agar Base, Oxoid, Wesel, Germany) and on ChromID *C. difficile* agar (bioMérieux, Nürtingen, Germany).

Plates and enrichment cultures were incubated under anaerobic conditions at 37 °C, plates for 2 to 3 days and enrichment cultures for 7 to 10 days. For spore selection, 1 mL of the enrichment culture was mixed with 1 mL of ethanol and incubated for 30 min to 1 h at room temperature. The mixture was centrifuged at 5.000× *g* for 10 min, the supernatant removed and the pellet resuspended in 200 µL NaCl (0.85%). Aliquots of 100 µL were plated onto CDMN and on ChromID *C. difficile* plates and incubated anaerobically at 37 °C for 2 to 3 days.

One morphologically suspicious colony (CDMN: irregular margins, branched and off-white colour; CDIF: irregular margins, grey or black colour) per sample was selected and cultivated. For species confirmation, DNA isolation of blood agar cultures of 48 h was performed with the DNeasy Blood and Tissue kit (Qiagen, Hilden, Germany), followed by a PCR of the *cdd3* gene with primers Tim 6 (5′-TCCAATATAATAAATTAGCATTCCA-3′) and Struppi 6 (5′-GGCTATTACACGTAATCCAGATA-3′) [51]. PCR was performed as described in Table 6.

### 4.3. Molecular Characterisation

PCR amplification of toxin genes was performed to detect the toxin genes *tcdA*, *tcdB*, *cdtA* and *cdtB* (Table 6), using the following primer pairs. The primers NK 2 (5′-CCCAATAGAAGATTCAATATTAAGCTT-3′) and NK 3 (5′-GGAAGAAAAGAACTTCTGGCTCACTCAGGT-3′) served to amplify the non-repetitive region of *tcdA,* NK 9 (5′-CCACCAGCTGCAGCCATA-3′) and NK 11 (5′-TGATGCTAATAATGAATCTAAAATGGTAAC-3′) the repetitive region of *tcdA*. For *tcdB*, the primer pair NK 104 (5′-GTGTAGCAATGAAAGTCCAAGTTTACGC-3′) and NK 105 (5′-CACTTAGCTCTTTGATTGCTGCACCT-3′) was used [52,53]. For the binary toxin, primers cdtApos (5′-TGAACCTGGAAAAGGTGATG-3′) and cdtArev (5′-AGGATTATTTACTGGACCATTTG-3′) were used to amplify *cdtA*; cdtBpos (5′-CTTAATGCAAGTAAATACTGAG-3′) and cdtBrev (5′-AACGGATCTCTTGCTTCAGTC-3′) for *cdtB* [54].

PCR ribotyping was performed with capillary gel electrophoresis according to a modified version of the protocol of Indra et al. [55]. Primers 16S 6FAM (5′-6FAM-GTGCGGCTGGATCACCTCCT-3′) and 23S (5′-PET-CCCTGCACCCTTAATAACTTGACC-3′) were used. DNA was diluted to 1 ng/µL and 1 µL was added to 24 µL of a PCR-mix containing 0.2 µL (5 U/µL) DreamTaq DNA polymerase (Thermo Fisher Scientific, Waltham, MA USA), 1 µL of each primer (working dilution of 10 pmol/µL), 1 µL of 10 mM dNTP-Mix (Carl Roth, Karlsruhe, Germany), 1 µL MgCl_2_ (25 mM) (Qiagen, Hilden, Germany), 2.5 µL DreamTaq buffer (Thermo Fisher Scientific, Waltham, MA USA) and 17.3 µL water.

PCR was performed as follows: 95 °C (2 min) for initial denaturation; 30 cycles of 95 °C (30 s) for denaturation, 52 °C (30 s) for annealing, 72 °C (2 min) for elongation; 72 °C (8 min) for the final extension. PCR products were diluted 1:75 with water and ribotyping was performed by capillary electrophoresis followed by a web-based analysis. The fragment separation was performed with a SeqStudio Genetic Analyzer, SeqStudio™ Cartridge v2 (POP-1 polymer, four capillaries, 28 cm capillary length) and GeneScan™ 600 LIZ™ Size Standard v2.0. The conditions of separation were defined via FragAnalysis Run Module. The fragment length results were converted into a suitable data format and analysed with the publicly accessible WEBRIBO database to assign a ribotype [56].

### 4.4. Whole Genome Sequencing

WGS was performed with the Illumina^®^ Miseq™ platform. Bioinformatic analysis of WGS data was performed with the pipeline WGSBAC (version 2.1.0) with following modules [13]: FastQC (v. 0.11.9) for quality control [57], shovill (v. 1.0.4) for de-novo genome assembly [58], QUAST (v. 5.0.2) for quality assessment of the genome assemblies [59], prokka (v. 1.14.6) for annotation [60,61], kraken2 (v. 2.1.2) for species confirmation [62], and snippy (v. 4.6.0) for SNP analysis [63]. The genome of *C. difficile* strain 630 (GenBank accession numbers NC_009089 and NC_008226) was used as reference. All tools were used with default settings.

cgMLST analysis was done with Ridom SeqSphere+ ™(v. 8.2.0) [16]. Antimicrobial resistance genes were searched using BLASTn as implemented in ABRicate v0.8 using the Bacterial Antimicrobial Resistance Reference Gene Database (NCBI BioProject PRJNA313047) [64].

Geneious Prime^®^ 2021.0.1 was used for detection of amino acid substitutions in GyrA and GyrB. Briefly, sequences of both genes were mapped to *gyrA* and *gyrB* of reference strain *C. difficile* 630 (GenBank accession number NZ_009089). Variations and SNPs were searched for, and amino acid changes in the deduced corresponding proteins were documented.

### 4.5. Antimicrobial Susceptibility Testing

For each isolate, the MIC for AMP, PEN, AMC, MEP, TET, CHL, ERY, CLI, MOX, CIP, MEZ, VAN and LZD was determined by Etest^®^ (bioMérieux, Nürtingen, Germany). For quality control, *Clostridioides* (*Clostridium*) *difficile* ATCC^®^ 700057 was included. As there are no quality control ranges provided by CLSI for CHL and TET for *C. difficile* ATCC^®^ 700057, *Bacterioides fragilis* ATCC^®^ 25285 was used additionally as quality control for these antimicrobial agents [19]. For ERY and CIP, no quality control ranges were available.

Etest^®^ was performed following the instructions for anaerobes (manual available as package insert online [17]) with the following adaptation: the inoculum was prepared by inoculating colonies of a 24 h old lawn culture grown on a Schaedler agar plate (Oxoid, Wesel, Germany) in 0.5 mL NaCl (0.85%).

For the classification of the results as susceptible/intermediate/resistant respectively WT or NWT (see Table 5), the MICs obtained were rounded up to the next two-fold standard dilution if they were in between two steps as indicated by the Etest^®^ instructions [17].

The breakpoints used according to CLSI and EUCAST as well as the epidemiological cut-off values (ECOFF) from EUCAST can be found in Table 5 [18,19,20]. As there is no official breakpoint nor an ECOFF for LZD, no classification in susceptible/resistant or WT/NWT was performed.

## 5. Conclusions

Isolation of *C. difficile* is described for the first time in the environment of South American camelids. The isolates were assigned to different RTs, all previously detected in humans. All isolates are toxinogenic and, therefore, are potentially pathogenic for humans. A hypervirulent, multidrug-resistant RT 078 isolate was isolated. All isolates showed resistance to antimicrobial agents or were classified as NWT. As the bacterial load of these samples was low, *C. difficile* carriage by SAC does not appear to pose a high risk for conferring CA-CDI. Nevertheless, SAC must not be ignored as a reservoir and potential source for transmission of *C. difficile* contributing to the complex epidemiology of these zoonotic bacteria.

## Figures and Tables

**Figure 1 antibiotics-12-00086-f001:**
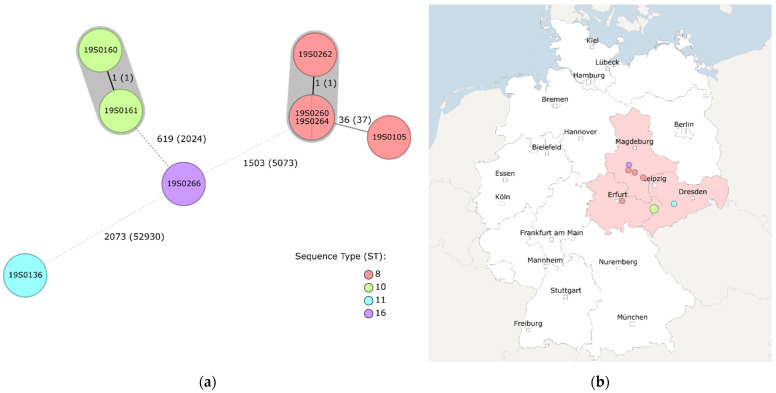
(**a**) Results of core genome multi locus sequence typing (cgMLST) of *C. difficile* isolates. Numbers next to the branches indicate numbers of allelic differences in the core genome. In brackets, the number of SNPs within the core genome is indicated. cgMLST classified the eight isolates into five genetic groups. Three isolates were singletons, while the other five isolates were grouped into two clusters (marked in grey) with two and three isolates, respectively. Each node represents a cgMLST sequence type (cgST) which was coloured according to their ST (classical MLST) classification. (**b**) Geographic origin (according to postal code) of the corresponding samples to the isolates. 19S0160 and 19S0161 (green; RT 015) originated from the same husbandry. RT 002/2 isolates (19S0260, 19S0162, 19S0264; red) originated from husbandries in Saxony-Anhalt which were close to each other, as did 19S0266 (purple; RT 029). 19S0136 (blue; RT 078) originated from a husbandry in Saxony. Figures were created with Ridom™ SeqSphere+ (v. 8.2.0) [16].

**Table 1 antibiotics-12-00086-t001:** Origin and molecular characteristics of isolates found in South American camelids (SAC).

Isolate	Farm-ID	Federal State	Ribotype (RT)	WEBRIBO-ID	Sequence Type (ST)	Toxin A	Toxin B	Binary Toxin
						*tcdA*	*tcdB*	*cdtA*/*cdtB*
19S0105	008	Thuringia	AI-75	PR28224	8	+	+	−/−
19S0136	016	Saxony	078	PR28225	11	+	+	+/+
19S0160	018	Saxony	015	PR26350	10	+	+	−/−
19S0161	018	Saxony	015	PR28235	10	+	+	−/−
19S0260	039	Saxony-Anhalt	002/2	PR28226	8	+	+	−/−
19S0262	040	Saxony-Anhalt	002/2	PR28226	8	+	+	−/−
19S0264	041	Saxony-Anhalt	002/2	PR26352	8	+	+	−/−
19S0266	042	Saxony-Anhalt	029	PR28237	16	+	+	−/−

**Table 2 antibiotics-12-00086-t002:** Pairwise single nucleotide polymorphisms (SNPs) between the eight *Clostridioides* (*C*.) *difficile* isolates.

Isolate	19S0105	19S0136	19S0160	19S0161	19S0260	19S0262	19S0264	19S0266	CD630
**19S0105**	0	91994	9286	9286	116	116	121	9306	9147
**19S0136**	91994	0	91623	91623	91999	91999	91996	91714	92383
**19S0160**	9286	91623	0	6	9310	9310	9307	3955	10527
**19S0161**	9286	91623	6	0	9310	9310	9307	3953	10531
**19S0260**	116	91999	9310	9310	0	0	5	9330	9173
**19S0262**	116	91999	9310	9310	0	0	5	9330	9173
**19S0264**	121	91996	9307	9307	5	5	0	9327	9168
**19S0266**	9306	91714	3955	3953	9330	9330	9327	0	10484
**CD630**	9147	92383	10527	10531	9173	9173	9168	10484	0

CD630: *C. difficile* strain 630 (GenBank accession numbers NC_009089 and NC_008226), reference strain.

**Table 3 antibiotics-12-00086-t003:** Resistance genes identified in the *C. difficile* isolates from SAC.

Isolate	Resistance Genes
19S0105	*bla*_CDD_-1; *vanG*; *vanR*; *vanS*; *vanT*; *vanZ1*
19S0136	*aadE*; *bla*_CDD_-1; *tet*(40); *tet*(M); *vanZ1*
19S0160	*bla*_CDD_-1; *vanG*; *vanR*; *vanS*; *vanT*; *vanZ1*
19S0161	*bla*_CDD_-1; *vanG*; *vanR*; *vanS*; *vanT*; *vanZ1*
19S0260	*bla*_CDD_-1; *vanG*; *vanR*; *vanS*; *vanT*; *vanZ1*
19S0262	*bla*_CDD_-1; *vanG*; *vanR*; *vanS*; *vanT*; *vanZ1*
19S0264	*bla*_CDD_-1; *vanG*; *vanR*; *vanS*; *vanT*; *vanZ1*
19S0266	*bla*_CDD_-1; *vanG*; *vanR*; *vanS*; *vanT*; *vanZ1*

**Table 4 antibiotics-12-00086-t004:** Amino acid substitutions in gyrase proteins in *C. difficile* isolates from SAC.

Isolate	Protein	Amino Acid Substitutions
19S0136	GyrA	Thr-82-Val * Lys-413-Asn
	GyrB	Gln-160-His Ser-366 ^#^-Val Ser-416-Ala *
19S0160	GyrB	Val-130-Ile
19S0161	GyrB	Val-130-Ile

* Amino acid substitutions that were reported to confer fluoroquinolone resistance [10]. ^#^ Another amino acid substitution at this position (Ser-366-Ala) was reported to confer fluoroquinolone resistance [10].

**Table 5 antibiotics-12-00086-t005:** Results of antimicrobial susceptibility testing (minimum inhibitory concentration, in mg/L) for *C. difficile* isolates from SAC.

Isolate (RT)	Antimicrobials	MEZ	VAN	MOX	CIP	AMP	PEN	AMC	MEP	ERY	CLI	CHL	TET	LZD
	*BP/ECOFF (mg/L)*	>2 ^a^	>2 ^a^	≤2/≥8 ^b^	32 ^c^	≤0.5/≥2 ^b^	≤0.5/≥2 ^b^	≤4/2/≥16/8 ^b^	≤4/≥16 ^b^	4 ^c^	≤2/≥8 ^b^	≤8/≥32 ^b^	≤4/≥16 ^b^	-
19S0105 (RT AI-75)		0.19	0.75	1.5	>32	1.5	1.5	0.75	1.5	2	8	12	0.38	3
19S0136 (RT 078)		0.125	0.75	>32	>32	0.75	1.5	0.38	0.75	>256	2	6	16	3
19S0160 (RT 015)		0.38	0.75	1.5	>32	1	1	1	1.5	2	12	6	0.38	6
19S0161 (RT 015)		0.125	0.75	1	>32	1.5	1	0.75	1.5	2	12	6	0.38	4
19S0260 (RT 002/2)		0.38	1	1.5	>32	1.5	1.5	0.5	1.5	2	8	8	0.19	3
19S0262 (RT 002/2)		0.38	0.75	1.5	>32	1.5	1.5	0.75	1.5	2	8	8	0.25	3
19S0264 (RT 002/2)		0.38	0.75	1.5	>32	1	1	0.5	1	2	12	8	0.094	3
19S0266 (RT 029)		0.38	1	1	>32	3	4	0.75	1.5	2	6	8	0.5	8

RT: Ribotype, BP: Breakpoints, ECOFF: Epidemiological cut-off values. MEZ: metronidazole, VAN: vancomycin, MOX: moxifloxacin, CIP: ciprofloxacin, AMP: ampicillin, PEN: penicillin, AMC: amoxicillin-clavulanate, MEP: meropenem, ERY: erythromycin, CLI: clindamycin, CHL: chloramphenicol, TET: tetracycline, LZD: linezolid. Etest^®^ provides a continuous scale for the determination of MIC values. Therefore, results can be obtained that are in-between the conventional two-fold dilutions. For interpretation, results were rounded to the next two-fold standard dilution if they were in between two steps as indicated by the instructions for Etest^®^ [17]: green: susceptible or wild type, yellow: intermediate, red: resistant or non wild type. ^a^ Breakpoints according to EUCAST [18]. ^b^ Breakpoints according to CLSI; first values indicate breakpoints for susceptible isolates, second values indicate breakpoints for resistant isolates [19]. ^c^ Epidemiological cut-off values from EUCAST [20].

**Table 6 antibiotics-12-00086-t006:** PCR-protocols for *C. difficile* species confirmation and toxin gene detection.

PCR-MIX and -Programme	Test Components	Working Concentration	*cdd3*	*tcdA*—Non-Repetitive Region	*tcdA*—Repetitive Region	*tcdB*	*cdtA*	*cdtB*
PCR-Mix	DNA	As obtained by DNA isolation	2 µL	2 µL	2 µL	2 µL	2 µL	2 µL
	DreamTaq Buffer (Thermo Fisher Scientific, Waltham, MA USA)	10 x	2.5 µL	2.5 µL	2.5 µL	2.5 µL	2.5 µL	2.5 µL
	dNTP-Mix (Carl Roth, Karlsruhe, Germany)	10 mM	0.5 µL	1 µL	0.5 µL	1 µL	0.5 µL	0.5 µL
	Primer 1 ^a^	10 pmol/µL	0.5 µL	1 µL	1 µL	1 µL	1 µL	1 µL
	Primer 2 ^a^	10 pmol/µL	0.5 µL	1 µL	1 µL	1 µL	1 µL	1 µL
	DreamTaq DNA Polymerase (Thermo Fisher Scientific, Waltham, MA USA)	5 U/µL	0.1 µL	0.2 µL	0.1 µL	0.2 µL	0.1 µL	0.1 µL
	MgCl_2_ (Qiagen, Hilden, Germany)	25 mM	1 µL	1 µL	1 µL	1 µL	1 µL	1 µL
	H_2_O		17.9 µL	16.3 µL	16.9 µL	16.3 µL	16.9 µL	16.9 µL
PCR-Programme	Cycles		35	35	35	35	35	35
	Initial Denaturation		2 min, 95 °C	2 min, 95 °C	2 min, 95 °C	2 min, 95 °C	2 min, 95 °C	2 min, 95 °C
	Denaturation		30 s, 95 °C	30 s, 95 °C	30 s, 95 °C	30 s, 95 °C	30 s, 95 °C	30 s, 95 °C
	Annealing		45 s, 50 °C	30 s, 55 °C	30 s, 62 °C	30 s, 55 °C	45 s, 52 °C	45 s, 52 °C
	Elongation		1 min, 72 °C	30 s, 72 °C	1,5 min 72 °C	30 s, 72 °C	40 s, 72 °C	40 s, 72 °C
	Finale Extension		8 min, 72 °C	5 min, 72 °C	8 min, 72 °C	5 min, 72 °C	8 min, 72 °C	8 min, 72 °C

^a^ The primers used are mentioned in the text.

## Data Availability

Publicly available datasets were analysed in this study. This data can be found here: https://www.ncbi.nlm.nih.gov/bioproject/902649 (accessed on 1 December 2022). BioSample accession numbers: SAMN31765301 (19S0105), SAMN31765302 (19S0136), SAMN31765303 (19S0160), SAMN31765304 (19S0161), SAMN31765305 (19S0260), SAMN31765306 (19S0262), SAMN31765307 (19S0264), SAMN31765308 (19S0266).

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
