# Peer review of "Clostridioides difficile in South American Camelids in Germany: First Insights into Molecular and Genetic Characteristics and Antimicrobial Resistance"

_antibiotics, 2023, doi:10.3390/antibiotics12010086_

Round 1
Reviewer 1 Report
Clostridioides (C.) difficile as an enteric pathogen can cause zoonotic transmission. This study investigates the prevalence, molecular characteristics and antimicrobial resistances of C. difficile in South American camelid (SAC) with increasing popularity and frequent contacting with humans.
As authors mentioned “All isolates were obtained only from enrichment cultures and belonged to five different PCR ribotypes (RT):” Could the different characteristics among those RTs be briefly described?
The samples of this text are very limited. Could authors do a mini review about others associated research which make your findings more stable?
This text did the analysis of single nucleotide polymorphisms (SNPs). But the meaning of this results needs to be explained simultaneously.
Reviewer 2 Report
Authors have addressed an important topic and is of a general interest in present time. Experiments are well designed as well as appropriately executed. Just a minor comments, author should address to improve MS furthter
1. Authors need to recheck presentation of Table 5, especially attributing status as to Sensitive, Intermediate and Resistant as per EUCAST/CLSI. They must ensure color coding is exactly matching.
2. Choice of traits do not look in the light of Extended Spectrum beta-lactamases classification by Ambler. Although its not must, however it would be better if authors can explain in brief rationale of choosing certain bla traits and not classifying isolated as per Ambler.
Best
Arun
